# On the Effectiveness of Quantum Chemistry Pre-training for Pharmacological Property Prediction

Arun Raja [1]   Hongtao Zhao [2]   Christian Tyrchan [2]   Eva Nittinger [2]   Michael M. Bronstein [1]
Charlotte M. Deane [1]   Garrett M. Morris [1]

## Abstract

In principle, quantum chemistry allows us to quantify all electronic and geometric properties of molecules and their interactions. Thus, incorporating pre-calculated quantum mechanical properties into deep learning models could improve their ability to predict important pharmacological properties of small molecules and potential drugs. However, this opportunity has been under-exploited in the recent wave of AI-driven drug discovery. We show that pre-training Equivariant Graph Neural Network (EGNN) models to predict atom-centered partial charges, that have been pre-calculated using quantum mechanical methods, we can obtain more accurate models to predict absorption, distribution, metabolism, excretion, and toxicological (ADMET) properties. We compared the performance of quantum chemistry pre-training against non-quantum mechanics-based pre-training and with no pre-training at all, and found quantum chemistry pre-training to produce the most accurate models for lipophilicity, blood-brain barrier penetration, metabolism by CYP2D6, and toxicity; and very similar performance to non-pre-training models for the much more challenging task of hepatocyte clearance prediction. By using our quantum chemistry-based pre-training to predict both atom-level and molecule-level properties, we obtain richer representations of the molecules than without pre-training, helping our models to learn from the underlying physics and chemistry.

[1]Department of Statistics, University of Oxford [2]Medicinal Chemistry, Research and Early Development, Respiratory and Immunology (RI), BioPharmaceuticals RD, AstraZeneca, 43183, Gothenburg, Sweden. Correspondence to: Arun Raja <arun.raja@dtc.ox.ac.uk>.

Machine Learning for Life and Material Science: From Theory to Industry applications Workshop at the $41^{st}$ International Conference on Machine Learning, Vienna, Austria. Copyright 2024 by the author(s).

## 1. Introduction

The study of quantititative-structure activity relationships (QSAR) is fundamental to medicinal chemistry and drug discovery. QSAR combines chemical intuition with computational and mathematical approaches to help in the search for molecules with desirable physicochemical and pharmacological properties. To find relevant structure-activity relationships it is necessary to use appropriate molecular descriptors. Recent advances in computational resources and algorithms has helped to boost the efficiency of performing quantum chemical calculations on molecules (Blunt et al., 2022). Such quantum chemical [1] descriptors of molecules are thus an additional source of featurization we could tap into. In principle, quantum chemistry allows us to quantify all electronic and geometric properties of molecules and their interactions (Griffiths & Schroeter, 2004).

Electronic descriptors can be derived directly from the molecular wavefunction. These molecule-level electronic descriptors can also partitioned at the level of atoms, in the form of atom-centered partial charges ($q$), or groups of atoms, which allows us to describe quantum chemistry at different scales. Previous research has demonstrated correlation between physicochemical properties calculated by quantum methods and experimentally-measured physiological properties, including Absorption, Distribution, Metabolism, Excretion, and Toxicity, or 'ADMET' (van Damme & Bultinck, 2008; del Amo, 2015; Silva-Junior et al., 2017). For instance, Mulliken partial charge-based descriptors such as the Mulliken charge separation on the hydrogens in a molecule $\left(q_{\mathrm{H}}^{\max} - q_{\mathrm{H}}^{\min}\right)$ and dipole moment $\left(\mu^2\right)$ are correlated with the degree of blood-brain barrier penetration, $\log \mathrm{BBB}$ (van Damme & Bultinck, 2008).

Thus, quantum-chemical methods can characterize a large amount of molecular and atomic properties including reactivity, conformation, and binding activity of a complete molecule and even molecular fragments. In turn, QSAR models built using these descriptors will contain information about the nature of intermolecular forces involved in

---

[1]In this work, the terms 'quantum chemical' and 'quantum mechanical' are used interchangeably.

determining the biological activity of the molecules (Cocchi et al., 1992). Quantum-chemical descriptors calculated by quantum chemical methods, unlike experimental measurements, lack aleatoric error, but can suffer from systematic errors that can be attributed to the inherent approximations made by these methods (Wang et al., 2021), such as the linear combination of atomic orbitals (LCAO) approximation. This systematic error, however, is considered to be approximately constant throughout a chemical series and can thus be neglected (Berkoff et al., 1976).

Considerable work has been carried out in the space of molecular representation learning using both molecular graphs and 3D geometries for ADMET property prediction— for example, UniMol (Zhou et al., 2023), MolCLR (Wang et al., 2022), GraphMVP (Liu et al., 2022) and GEM (Fang et al., 2021). UniMol uses 3D atomic coordinate prediction and atom masking as pre-training strategies. MolCLR uses contrastive learning on molecular graphs for the pre-training procedure. GraphMVP also employs contrastive learning between graphs and 3D geometries. GEM uses bond angles and bond lengths as additional 3D information. In this recent wave of AI-driven drug discovery research, however, the use of quantum chemical information or knowledge has been under-exploited in ADMET property prediction.

A key advantage of using 3D geometry—that is, the interplay between the geometry and electronic structure of molecules in determining their properties and intermolecular interactions—is not realized when quantum chemical information is ignored. In those cases where both 3D and quantum chemical information have been used together in pre-training, the resulting models have been used to predict other quantum chemical properties such as dipole moments (Wang et al., 2023). We hypothesized that quantum chemistry pre-training may also be effective for ADMET property prediction. We performed a wide range of experiments exploring different types of pre-training and fine-tuning on datasets spanning a variety of ADMET properties. Our aim was to: (1) motivate the use of quantum chemical information in drug property prediction; (2) identify potential pitfalls; and (3) call for larger drug-related quantum chemistry datasets to be released. An overview of our quantum chemistry transfer learning pipeline is given in Figure **??**, and elaborated below. We compare (i) the effect of pre-training versus no pretraining; (ii) the use of 2D graphs versus equivariant 3D graphs; and (iii) three ways of calculating atom-centred partial charges: non-quantum 'topological' partial charges (Gasteiger); and two quantum chemical charge calculation methods (Mulliken and Löwdin).

## 2. Methodology

We used the Equivariant Graph Neural Network or EGNN (Satorras et al., 2022) to encode each molecule's 3D ge-

ometry; and the GraphSAGE (Hamilton et al., 2018) graph neural network to encode the molecule as a 2D graph. We used the same model architectures for both pre-training and downstream fine-tuning to investigate the effectiveness of quantum chemistry pre-training. We used the EGNN and GraphSAGE models for ADMET property prediction in three training regimes:

- *No pre-training*, *i.e.*, direct prediction of each ADMET property;

- *Non-quantum chemical pre-training* to predict 'topological' Gasteiger partial charges (Gasteiger & Marsili, 1980) using GraphSAGE; and

- *Quantum chemical pre-training* to predict: (i) quantum mechanical partial charges, namely Mulliken (Mulliken, 1955) or Löwdin (Löwdin, 1970) charges; and (ii) the HOMO-LUMO gap of the molecule's highest occupied and lowest unoccupied molecular orbital (a measure of its chemical reactivity).

In the last two cases, the resulting molecular embedding is used as input to a fine-tuning phase to predict the desired ADMET property.

The electron distribution in a molecule allows us to understand the tendency for certain intermolecular interactions to occur. One way to approximate the electronic distribution in a molecule at an atomic level is via atom-centred partial charges. Thus we chose to pre-train the models on partial charges to attain node or atom-level embeddings. There are two distinct classes of partial charges: non-quantum mechanical (QM) and QM-based partial charges.

### 2.1. Control: Pre-training on Gasteiger Partial Charges

The Partial Equalization of Orbital Electronegativity (PEOE) method by Gasteiger & Marsili (1980) assumes that the electronegativity, $\chi_v$, of an atom type, $v$, is a quadratic function of the atomic partial charge:

$$\chi_v = a_v + b_v \left( q_v \right) + c_v \left( q_v \right)^2 \tag{1}$$

where $q_v$ is the partial charge, and $a_v, b_v$, and $c_v$ are coefficients to be optimized. According to Mulliken (1934), the electronegativity, $\chi_v$, of an atom is related to its ionization potential, $I_v$, and its electron affinity, $E_v$, as follows:

$$\chi_v = \frac{1}{2} \left( I_v + E_v \right) \tag{2}$$

Next, the partial charges are updated using an iterative process of calculating charge transfers between bonded atoms, until convergence. Initially, all atoms are assigned charges

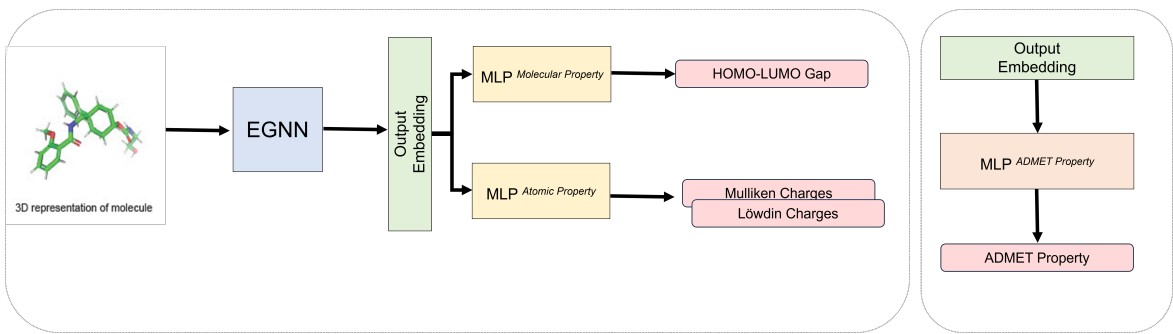

*Figure 1.* An overview of quantum chemistry transfer learning. *Left:* We pre-train an EGNN to predict both atomic and molecular quantum chemical properties to create a rich output embedding, that can subsequently be used to predict ADMET properties (*right*).

based on their atom type. At each step, charge is transferred from atoms of lower electronegativity $\chi_v$ to bonded atoms of higher electronegativity $\chi_{v'}$ thus:

$$\Delta Q_{v \to v'}^{(n)} = \frac{\chi_{v'}^{(n-1)} - \chi_v^{(n-1)}}{a_v + b_v + c_v} f_{vv'}^n \qquad (3)$$

where $f$ is a damping factor and is set to 0.5 in Gasteiger & Marsili (1980). Therefore, Gasteiger partial charges are based on the molecule's topology (not 3D conformation) and its atoms' electronegativity values, which are in turn related to their ionization potential and electron affinity, *i.e.*, experimentally measured physicochemical properties. Using a model pre-trained on Gasteiger partial charges which are non-quantum mechanical in nature acts as a control experiment to understand if QM pre-training helps.

## 2.2. Pre-Training on Quantum Chemical Partial Charges and Molecular Properties

QM-based partial charges are calculated by partitioning the molecular wavefunction into atom-level contributions. Mulliken and Löwdin partial charges are derived by distributing the electrons amongst the atoms according to the degree to which different atomic orbital basis functions contribute to the molecular wavefunction. This partitioning scheme is known as 'population analysis' (Mulliken, 1955). The electronic population is defined as follows:

$$
\begin{aligned}
N &= \sum_{j}^{\text{electrons}} \int \psi_j(\mathbf{r}_j) \psi_j(\mathbf{r}_j) \, d\mathbf{r}_j \\
&= \sum_{j}^{\text{electrons}} \sum_{r,s} \int c_{jr} \varphi_r(\mathbf{r}_j) c_{js} \varphi_s(\mathbf{r}_j) \, d\mathbf{r}_j \qquad (4) \\
&= \sum_{j}^{\text{electrons}} \left( \sum_r c_{jr}^2 + \sum_{r \neq s} c_{jr} c_{js} S_{rs} \right)
\end{aligned}
$$

where $r$ and $s$ index the atomic orbital basis functions, $\varphi_r$ and $\varphi_s$; $c_{jr}$ is the coefficient of basis function, $\varphi_r$, in the molecular orbital, $\psi_j$; and $S_{rs}$ is the overlap matrix element[2].

In Eqn. 4, the total number of electrons is represented by two sums: one including only squares of single atomic orbital basis functions ($\sum_r c_{jr}^2$), and the other including products of two different atomic orbital basis functions, $r$ and $s$ ($\sum_{r \neq s} c_{jr} c_{js} S_{rs}$). The first summation covers electrons 'residing' on a single atom, whereas the second summation includes all electrons shared between basis functions. Mulliken (1955) suggested that overlapping orbital electron be divided evenly between the two parent atoms' basis functions $r$ and $s$. On the other hand, Löwdin (1970) performed orthogonalization on the atomic orbital basis sets before doing population analysis. For this reason, Löwdin partial charges are said to be more 'stable' than Mulliken charges.

In addition to predicting QM-based partial charges derived by partitioning the molecular wavefunction, we hypothesized that it could also be beneficial to simultaneously pre-train on a molecular QM property such as the HOMO-LUMO gap. Prior work in GNN-based pre-training shows that even richer molecular representations can be realized by jointly learning node-level *and* graph-level embeddings (Hu et al., 2020). We next discuss the experimental details of our pre-training and fine-tuning phases.

## 3. Experimental Design

To pre-train on quantum chemical data, we separately used the QM9 (Ramakrishnan et al., 2014), then the QMugs (Isert et al., 2021) datasets. QM9 contains 3D structures and QM properties calculated at the B3LYP/6-31G(2df,p) level of theory, and consists of 133,885 small organic molecules containing up to nine C, N, O, and/or F heavy atoms (as

---

[2]For further details please refer to Cramer (2002)

well as H)—a subset of the GDB-17 set of 166 billion organic molecules. We used the HOMO-LUMO gap and the per-atom Mulliken partial charges as labels for pre-training. QMugs, on the other hand, contains ~665,000 drug-like molecules. Ten elements are represented in QMugs molecules: C, H, N, O, S, P, F, Cl, Br, and/or I. QMugs provides both Mulliken and Löwdin partial charges and molecular properties at two levels of theory: $\omega$B97X-D/def2-SVP and the semiempirical GFN2-xTB. We the charges and properties calculated at the $\omega$B97X-D/def2-SVP level only [3]. In QMugs, a maximum of three conformations are provided for each molecule. The lowest energy conformation of each molecule was used. For the non-QM pre-training control case, we generated Gasteiger partial charges for molecules in QM9 using RDKit (Landrum) and set the number of iterations to 12.

To pre-train on Gasteiger partial charges, the GraphSAGE model with 2D molecular graphs was used as Gasteiger charges are topological and do not depend on a molecule's conformation. The EGNN was used for predicting the quantum mechanically calculated properties: Mulliken and Löwdin partial charges and the HOMO-LUMO gap, as these properties depend on the 3D structure of the molecule. The EGNN used is E(3)-invariant as these properties are invariant to translations, rotations, and reflections of the atom positions. The results for the pre-training phase are reported in the Appendix (Table 4).

### 3.1. Fine-Tuning Datasets for Property Prediction

We used the following datasets for ADMET property prediction and their corresponding scaffold splits in Therapeutic Data Commons (TDC) (Huang et al., 2021). A dataset for each of the ADMET properties was selected as exemplars, to understand the effectiveness of quantum chemistry pre-training on a wide range of tasks:

- Lipophilicity (regression): the ability of a drug to dissolve in a lipid environment, as measured by the octanol/water distribution coefficient ($\log_D$ at pH 7.4) (Wu et al., 2018);

- Blood-Brain Barrier Penetration, BBBP (binary classification): whether a drug penetrates the blood-brain barrier (Wu et al., 2018);

- Substrate of Cytochrome P450$_{2D6}$, CYP2D6 (binary classification): cytochrome P450$_{2D6}$ is primarily expressed in the liver; this dataset indicates if a molecule is a substrate of CYP2D6 (Carbon-Mangels & Hutter, 2011);

- Clearance-Hepatocyte (regression): the rate of plasma cleared of a drug (Liu et al., 2007); and

- Acute Toxicity, $LD_{50}$ (regression): the most conservative dose that kills half the population tested, measured in $\log(1/(mol/kg))$ (Zhu et al., 2009).

TDC provides the molecules as SMILES strings for each dataset. These were converted to molecular graphs for predicting their labels using GraphSAGE. For 3D geometries, the single lowest energy conformer was generated using ETKDG (Riniker & Landrum, 2015) with Merck Molecular Force Field 94 (Halgren, 1996) optimization in RDKit. All models were trained, validated and tested on the downstream datasetswith 10 different random number generator seeds, for which the means and standard deviations of the performance metrics are reported.

### 3.2. Transfer Learning

Our goal was to build a transfer learning approach for pharmacological property (ADMET) prediction using quantum chemistry pre-training. Pharmacological properties are unlike quantum chemical properties for the following reasons:

- The 'downstream' properties in Section 3.1 are experimentally measured quantities, unlike calculated quantum chemical quantities which are obtained by using computational methods that are based on varying levels of theory (such as $\omega$B97X-D/def2-SVP).

- The different sources of these pharmacological properties mean different kinds of errors can be associated with them. Random error may be caused by different humans performing the assays for the downstream datasets; whereas for quantum properties, the sources of error are systematic and caused by assumptions made in the algorithms (which can be neglected as it would be present in all molecules in that dataset). User error is also possible when running calculations.

- It is not possible to calculate these downstream properties using the quantum chemical descriptors as there is no direct mathematical relationship between them (Karelson et al., 2010).

We thus treat pharmacological property prediction as an out-of-distribution (OOD) problem and chose the *linear probing* approach which is known to be better than fine-tuning the entire model for OOD scenarios (Kumar et al., 2022). Linear probing involves freezing lower layers to act as a 'feature extractor', then fine-tuning a head specific to the downstream task (see Figure 1, right). In this work, for both the EGNN and GraphSAGE, we froze the lower layers (before the multi-layer perceptron head for partial charge

---

[3]At the GFN2-xTB level, Löwdin partial charges are known not to be rotationally invariant (Bruhn et al., 2006), so we have avoided using these.

and/or HOMO-LUMO gap prediction) and used them as a frozen feature extractor. We used a randomly initialized multi-layer perceptron head where only the head is trained for the downstream task.

Kumar et al. (2022) also showed that fine-tuning the entire model after linear probing has better OOD performance compared than just using linear probing, and we have demonstrated this successfully for lipophilicity prediction (Table 3). However, we would like to emphasize that the focus of our work is to investigate the effectiveness of quantum chemistry pre-training and not the effectiveness of the different types of transfer learning approaches.

## 4. Results and Discussion

The performance metrics and units for the target datasets are listed in Tables 1 and 2. The QM9 pre-training consists of both non-QM-based pre-training to predict Gasteiger partial charges, and QM-based pre-training to predict Mulliken partial charges and HOMO-LUMO gap [4]. GraphSAGE pre-trained on Gasteiger charges results in worse performance on Lipohilicity ('Absorption'), BBB ('Distribution'), and CYP2D6 ('Metabolism') compared to its non-pre-trained counterparts. This suggests that electronegativity, the property from which Gasteiger partial charges are derived, is less relevant to the downstream task. Particularly, in the case of CYP2D6, the pre-trained GraphSAGE has an AU-ROC of $0.489 \pm 0.116$—very close to a random classifier considering the large standard deviation. This highlights that Gasteiger charge-pre-training is not relevant to studying metabolism. In the case of metabolism, a non-pre-trained EGNN also does not outperform a random classifier. However, there is a significant improvement when the QM-pre-trained EGNN models are used to predict substrate metabolism. Pre-training on *both* node and graph levels purportedly results in richer representations (Hu et al., 2020) as evidenced by them performing best for metabolism.

On the other hand, pre-training on both Mulliken charges and HOMO-LUMO gap falls short of the Mulliken charge-only pre-trained variant for BBB penetration classification. Here, the non-pre-trained EGNN performs better than the QM9 pre-trained variants. However, the QMugs variants perform best, suggesting that the QM9 pre-trained variants may be limited by their smaller pre-training dataset size (135k)—about five times smaller than QMugs (650k). Furthermore, for BBB, we note that the combined node and graph-level pre-trained variants perform slightly worse than just node-level pre-trained models. This may be due to a non-optimal pre-training target for the molecule level property, *i.e.*, the HOMO-LUMO gap. Other molecular quantum

properties like dipole moments may have proven to be more relevant to BBB penetration prediction. The continued exploration of a wider range of molecular quantum property prediction in the future (Section 5) will increase our understanding of which quantum properties are most relevant for a given ADMET property.

### 4.1. Pre-Training on QM9 versus QMugs

For lipophilicity and toxicity prediction, models pre-trained on partial charges and the HOMO-LUMO gap perform the best, supporting our hypothesis. However, for lipophilicity, the QMugs pre-trained model performs the best, whereas for toxicity the QM9 pre-trained model edges out the others. The best performance of QMugs pre-trained models might be attributed to their larger pre-training dataset size. On the other hand, QM9 is smaller and also contains fewer element types (half that of QMugs). This restricts models pre-trained on QM9 can only be used for molecules in the downstream datasets that have the same subset of elements, which in turn reduces the test data size significantly, making it easier to perform well [5]. In addition, the quantum properties in these datasets have been calculated using different levels of theory: B3LYP/6-31G(2df,p) in QM9 and $\omega$B97X-D/def2-SVP in QMugs which means different quantum observables are being considered in each method. For instance, B3LYP does not consider London dispersion effects (Bursch et al., 2022).

### 4.2. Pre-Training on Löwdin versus Mulliken Charges

Models pre-trained on Löwdin charges almost always outperform those pre-trained on Mulliken charges, except for BBB, although the performance is extremely similar (Tables 2). Pre-training on Löwdin charges may be advantageous given their orthogonalization step on the atomic orbital basis sets as explained in Section 2.2. However, both types of quantum charges belong to the same class, Class II, of partial charges as described by Cramer (2002) in that they partition the wavefunction arbitrarily. In the future, other types of partial charges, especially semi-empirical ones like CM1 (Storer et al., 1995) which fall within Class IV of Cramer (2002) can be explored.

## 5. Conclusion and Future Work

We have shown that quantum chemistry pre-training is effective for ADMET property prediction. Pre-training on both molecular and atomic labels such as HOMO-LUMO gap and partial charges, respectively, leads to better performance on most downstream tasks. We encourage the computational drug discovery community to start using quantum chemical

---

[4]HOMO-LUMO gap has been abbreviated as 'Gap' in the tables

[5]The corresponding downstream dataset sizes for atom-types found in QM9 and QMugs are given in Table 5.

| Type of pre-training | Downstream datasets | | | | |
|---|---|---|---|---|---|
| | Absorption | Distribution | Metabolism | Excretion | Toxicity |
| | Lipophilicity, AstraZeneca RMSE (logD units) ↓ | BBB AUROC ↑ | CYP2D6-Substrate AUROC ↑ | Clearance-Hepatocyte Spearman correlation coefficient ↑ | Acute Toxicity $LD_{50}$ RMSE (log[1/(mol/kg)] units)↓ |
| None - GraphSAGE | $0.867 \pm 0.052$ | $0.534 \pm 0.112$ | $0.573 \pm 0.066$ | $0.283 \pm 0.095$ | $0.798 \pm 0.053$ |
| None -EGNN | $0.767 \pm 0.069$ | $\mathbf{0.806 \pm 0.042}$ | $0.502 \pm 0.006$ | $\mathbf{0.432 \pm 0.094}$ | $0.802 \pm 0.055$ |
| Gasteiger - GraphSAGE | $0.881 \pm 0.042$ | $0.524 \pm 0.108$ | $0.489 \pm 0.116$ | $0.353 \pm 0.144$ | $0.771 \pm 0.077$ |
| Mulliken only - EGNN | $0.715 \pm 0.041$ | $0.743 \pm 0.176$ | $0.778 \pm 0.079$ | $0.417 \pm 0.130$ | $0.705 \pm 0.067$ |
| Mulliken + Gap - EGNN | $\mathbf{0.707 \pm 0.040}$ | $0.647 \pm 0.133$ | $\mathbf{0.878 \pm 0.069}$ | $0.410 \pm 0.176$ | $\mathbf{0.700 \pm 0.070}$ |

*Table 1.* A comparison of non-pre-trained models against QM9 pre-trained models (↑ higher the better, ↓ lower the better )

| Type of pre-training | Downstream datasets | | | | |
|---|---|---|---|---|---|
| | Absorption | Distribution | Metabolism | Excretion | Toxicity |
| | Lipophilicity, AstraZeneca RMSE (logD units) ↓ | BBB AUROC ↑ | CYP2D6-Substrate AUROC ↑ | Clearance-Hepatocyte Spearman correlation coefficient ↑ | Acute Toxicity $LD_{50}$ RMSE (log[1/(mol/kg)] units) ↓ |
| None - EGNN | $0.767 \pm 0.069$ | $0.806 \pm 0.042$ | $0.502 \pm 0.006$ | $\mathbf{0.432 \pm 0.094}$ | $0.802 \pm 0.055$ |
| Mulliken only - EGNN | $0.760 \pm 0.053$ | $\mathbf{0.864 \pm 0.032}$ | $\mathbf{0.771 \pm 0.067}$ | $0.413 \pm 0.071$ | $0.779 \pm 0.040$ |
| Löwdin only - EGNN | $0.724 \pm 0.047$ | $0.863 \pm 0.030$ | $0.766 \pm 0.059$ | $0.418 \pm 0.061$ | $0.779 \pm 0.039$ |
| Mulliken + Gap - EGNN | $0.731 \pm 0.037$ | $0.860 \pm 0.031$ | $0.769 \pm 0.068$ | $0.400 \pm 0.091$ | $0.781 \pm 0.042$ |
| Löwdin + Gap - EGNN | $\mathbf{0.688 \pm 0.039}$ | $0.861 \pm 0.029$ | $0.767 \pm 0.062$ | $0.424 \pm 0.110$ | $\mathbf{0.771 \pm 0.050}$ |

*Table 2.* A comparison of non-pre-trained models against QMugs pre-trained models (↑ higher the better, ↓ lower the better )

| TYPE OF PRE-TRAINING | LINEAR PROBING | LPFT |
|---|---|---|
| MULLIKEN ONLY | $0.760 \pm 0.053$ | $\mathbf{0.691 \pm 0.046}$ |
| LÖWDIN ONLY | $0.724 \pm 0.047$ | $\mathbf{0.643 \pm 0.057}$ |
| MULLIKEN + GAP | $0.731 \pm 0.037$ | $\mathbf{0.701 \pm 0.062}$ |
| LÖWDIN + GAP | $0.688 \pm 0.039$ | $\mathbf{0.599 \pm 0.051}$ |

*Table 3.* RMSE performance of linear probing then fine-tuning (LPFT) against just linear probing on the Lipophilicity dataset with pre-training on the QMugs dataset

descriptors and representations for pharmacological property prediction. Moreover, more quantum chemical data has to be generated for drug-like molecules. QMugs has certainly proved to be useful but much larger datasets with more types of elements and molecules will be beneficial for various drug discovery tasks. There are viable future directions one could take to further assess the effectiveness of quantum chemistry pre-training and make it more useful as discussed in Section 4—a wider range of molecular quantum properties like dipole moments and total energy could be used to understand which specific or combination of properties results in better downstream performance for a particular pharmacological property. A deeper investigation into why pre-training on Löwdin charges results in better downstream performance is necessary to choose the right class of partial charges for pre-training. With the recent advancements in quantum chemistry algorithms, computational hardware, and the evidence presented here showing the effectiveness of learning quantum chemistry, we hope that more work will be carried out to build on the progress in combining quantum chemistry with AI to accelerate drug discovery.

## Conflict of Interest

H.Z., E.N. and C.T. are employees of AstraZeneca and may own stock or stock options.

## Acknowledgements

A.R.'s PhD program is supported by the Agency for Science Technology and Research and the SABS R3 CDT program via the Engineering and Physical Sciences Research Council.

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

# A. Appendix

## A.1. Pre-training results

| Dataset | Partial charge type | Model | Partial charge loss (q) | Molecular charge loss (Q) | Total loss (Q+q) | Gap loss |
|---------|--------------------|-------|------------------------|--------------------------|------------------|----------|
| QM9 | Gasteiger | GraphSAGE | 6.981e-5 | 8.074e-3 | 8.144e-3 | N.A. |
|  | Mulliken | EGNN | 7.872e-5 | 3.997e-4 | 4.784e-4 | 3.063e-5 |
| QMugs | Mulliken | EGNN | 1.488e-4 | 1.788e-3 | 1.936e-3 | 3.909e-5 |
|  | Löwdin | EGNN | 3.790e-5 | 1.667e-3 | 1.705e-3 | 5.297e-5 |

*Table 4.* Pre-training results for QM9 and QMugs. The metric used was mean squared loss and the units are eV. HOMO-LUMO gap has been abbreviated as 'Gap' in the table

## A.2. Downstream dataset sizes

| Downstream dataset | Original #Samples | #Samples with atom-types found in QM9 | #Samples with atom-types found in QMugs |
|-------------------|------------------|--------------------------------------|---------------------------------------|
| Lipophilicity | 4200 | 2080 | 4192 |
| BBB | 2030 | 1184 | 2010 |
| CYP2D6-Substrate | 667 | 415 | 666 |
| Clearance-Hepatocyte | 1213 | 537 | 1209 |
| Acute Toxicity | 7385 | 4063 | 7282 |

*Table 5.* Downstream ADMET dataset sizes. QMugs contains 10 atom-types: C, H, N, O, S, P, F, Cl, Br, and/or I whereas QM9 contains 5 atom-types: C, H, N, O, and/or F

