# OpenReview forum: "On the Effectiveness of Quantum Chemistry Pre-training for Pharmacological Property Prediction"
_ICML.cc/2024/Workshop/ML4LMS — ML4LMS Poster_

### Official Review · Reviewer_9BK2 · 2024-06-11
**Review of 3D QM Pretraining**

**Rating:** 7
**Confidence:** 2

**Review:**

Positives:
1. This is one of the first papers to use a 3D GNN pretrained on QM data to do pharmacological property prediction.
2. There are some nice insights, such as electronegativity not being as relevant for absorption, distribution, and metabolism prediction.
3. There is good discussion around the different levels of QM theory and which might be useful for predicting which properties.

Concerns:
1. The authors use TDC, yet report results only on GraphSAGE, a very bad baseline.
2. AUROC is used for CYPD6 instead of AUPRC as in the baseline, so they don't follow the guidelines.
3. Some other papers that use QM should be cited, such as 3D infomax, GPS++, MolGPS.

Overall, the baseline table needs some work, but this is a good workshop paper with promising initial results and good perspectives.

---

### Official Review · Reviewer_zjmd · 2024-06-12

**Rating:** 6
**Confidence:** 4

**Review:**

**Summary**

This work demonstrates that pretraining on quantum mechanical properties like partial charges can help in various pharmacological property prediction tasks. Pre-training is performed with non-quantum based partial charges as well as quantum based partial charges.

**Positives**

- This work defines the pertaining and fine-tuning datasets very well based on the problem the motivation of their work. Pretraining on different (quantum and non-quantum charges) node level properties and graph level property. Finetuning on experimental ADMET data.
- Ablation studies include the impact of pertaining on each of the tasks for all fine-tuning datasets.

**Concerns**

- I understand that Gasteiger charges might not need 3D coordinates geometric information. It would still be good to have results with gasteiger charges with EGNN backbone to have a fair comparison on the impact of pertaining (on gasteiger charges compared to others).  I’d basically like to see “Gasteiger-EGNN” results in Table 1 and 2.